# Multi-omics investigation of spontaneous T2DM macaque emphasizes gut microbiota could up-regulate the absorption of excess palmitic acid in the T2DM progression

Xu Liu[1†], Yuchen Xie[1†], Shengzhi Yang[1], Cong Jiang[1], Ke Shang[1], Jinxia Luo[1,2], Lin Zhang[1], Gang Hu[3], Qinghua Liu[3], Bisong Yue[1], Zhenxin Fan[1], Zhanlong He[4], Jing Li[1]*

[1]Key Laboratory of Bio-resources and Eco-environment (Ministry of Education), College of Life Sciences, Sichuan University, Chengdu, China; [2]Sichuan Key Laboratory of Development and Application of Monkey Models for Human Major Disease, Sichuan, China; [3]SCU-SGHB Joint Laboratory on Non-human Primates Research, Meishan, China; [4]Institute of Medical Biology, Chinese Academy of Medical Sciences & Peking Union Medical College, Yunnan, China

*For correspondence:
ljtjf@126.com

†These authors contributed equally to this work

## eLife Assessment

This **important** work substantially advances our understanding of the interaction among gut microbiota, lipid metabolism, and the host in type 2 diabetes. The evidence supporting the claims of the authors is **convincing**. The work will be of interest to medical biologists working on microbiota and diabetes.

**Abstract** Although gut microbiota and lipid metabolites have been suggested to be closely associated with type 2 diabetes mellitus (T2DM), the interactions between gut microbiota, lipid metabolites, and the host in T2DM development remains unclear. Rhesus macaques may be the best animal model to investigate these relationships given their spontaneous development of T2DM. We identified eight spontaneous T2DM macaques and conducted a comprehensive study investigating the relationships using multi-omics sequencing technology. Our results from 16 S rRNA, metagenome, metabolome, and transcriptome analyses identified that gut microbiota imbalance, tryptophan metabolism and fatty acid β oxidation disorders, long-chain fatty acid (LCFA) accumulation, and inflammation occurred in T2DM macaques. We verified the accumulation of palmitic acid (PA) and activation of inflammation in T2DM macaques. Importantly, mice transplanted with spontaneous T2DM macaque fecal microbiota and fed a high PA diet developed prediabetes within 120 days. We determined that gut microbiota mediated the absorption of excess PA in the ileum, resulting in the accumulation of PA in the serum, consequently leading to T2DM in mice. In particular, we demonstrated that the specific microbiota composition was probably involved in the process. This study provides new insight into interactions between microbiota and metabolites and confirms causative effect of gut microbiota on T2DM development.

## Introduction

Diabetes mellitus is considered to be a refractory disease causing a significant socio-economic burden. T2DM is the dominant type of diabetes mellitus characterized by metabolic disorders, insulin resistance, and deficiency of insulin secretion (*American Diabetes Association, 1997*; *Zhang et al., 2021*). The pathogenesis of this chronic disease is complex and genetic and environmental factors, such as sugar and lipid intake, gut microbiota, many metabolites, and even air pollutants, contribute to its increase in prevalence (*Kahn et al., 2014*; *Kang et al., 2022*). Accumulating evidence has linked gut microbiota with T2DM development in a variety of ways. The gut microbiota can impact the integrity of the intestinal epithelial barrier, mediate insulin resistance, as well as regulate the function of mitochondria (*Qin et al., 2012*; *Zhao et al., 2018*; *Karlsson et al., 2013*). They can regulate local or systemic immunity and inflammation, which also contributes to the development of T2DM (*Tilg and Moschen, 2014*). Moreover, various gut microbial metabolites, such as short-chain fatty acids, bile acids, and tryptophan-derived metabolites, have been reported to be closely related to the pathogenesis of T2DM (*Qin et al., 2012*; *Zhao et al., 2018*; *Hendrikx and Schnabl, 2019*; *Qi et al., 2022*). However, the interactions between gut microbiota and its host with T2DM have not yet been fully characterized. In pathological conditions, the dysregulation of the host can lead to changes in gut microbiota composition. In turn, the microbiota plays a regulatory role to participate in the development of T2DM. Despite the complicated interactions between host and microbiota in the context of T2DM, some studies suggest antidiabetic interventions targeting the gut microbiota, such as fecal microbiota transplanting (FMT) can be applied as a clinical treatment of T2DM (*Wu et al., 2022*; *Ng et al., 2022*; *Ding et al., 2022*).

Meanwhile, dysfunction of lipid metabolism contributes to T2DM development by inducing lipotoxicity in humans and animal models (*Unger and Zhou, 2001*; *Poitout and Robertson, 2008*). The LCFAs are the principal lipid components naturally occurring in animal fats and vegetable oils, as well as the main metabolites of fat. LCFAs such as palmitic acid (PA, C16:0), palmitoleic acid (C16:1N7), and oleic acid (C18:1N9) are reported to have a strong association with T2DM. In the last few decades, there has been increasing evidence that frequent consumption of LCFAs contributes to metabolic diseases such as obesity and T2DM due to the high PA content (*Palomer et al., 2018*; *Milanski et al., 2009*). PA is a saturated fatty acid and its increase in serum is a significant contributing factor in T2DM development (*Fiehn et al., 2010*; *Li et al., 2009*; *Chandra et al., 2020*). The suggested mechanisms by which PA mediates T2DM include increasing diacylglycerol and ceramide synthesis (*Ertunc and Hotamisligil, 2016*), mitochondrial and endoplasmic reticulum stress (*Salvadó et al., 2015*; *Tumova et al., 2016*), and activation of pro-inflammatory pathways (*Cani et al., 2007*). Nevertheless, whether gut microbiota involving in PA- mediated T2DM and the interactions between gut microbiota and LCFA metabolites in T2DM development are still unclear.

Spontaneous development of T2DM in non-human primates (e.g. macaques) is highly similar to human T2DM, such as insulin resistance in early stages and later abnormal glucose tolerance and T2DM development follows the same pathological changes of pancreatic islets and complications. In fact, T2DM macaques avoid medication interference and environmental heterogeneity under controlled experimental conditions, and share key pathological features with humans, such as amyloidosis of pancreatic islets, which is absent in mouse models (*Birdwell et al., 2022*; *Westermark et al., 2011*), suggesting that T2DM macaques are the optimal animal model for simulating human T2DM and its complications (*Cefalu, 2006*). However, previous studies indicated that naturally occurring spontaneous T2DM macaques in captive populations were rare even if individuals were given a high lipid and high sugar diet (*Ji et al., 2012*; *Sun et al., 2020*). After nasally fed cynomolgus macaques with a high-fat dietary emulsion for 12 months, the macaques did not experience significant increases in fasting blood glucose and glycosylated hemoglobin (*Sun et al., 2020*). The controversial effects of high fat on the T2DM development are worthy of further investigation to better understand this complex disease.

Here, this study identified spontaneous development of T2DM in individuals (hereafter spontaneous T2DM macaques) from a large group of rhesus macaques. These spontaneous T2DM macaques have never been treated with anti-diabetic drugs and, therefore, provide valuable models for pathogenesis investigation of T2DM. Based on the macaque model, we used multi-omics techniques to address the interactions between gut microbiota, host gene expression, and fecal metabolites and the development of T2DM. Our results demonstrated that gut microbiota and LCFA metabolites played

**Table 1.** Physiological and biochemical parameters of Control and type 2 diabetes mellitus (T2DM) groups.

| Index | T2DM (n=8) | Control (n=8) |
|---|---|---|
| FPG (mmol/L) | 7.75±1 | 4.05±1.02[**] |
| FPI (μU/mL) | 17.97±8.50 | 6.45±2.56[**] |
| HOMA-IR | 6.24±3.06 | 1.20±0.65[**] |
| BMI | 14.03±7.71 | 14.30±1.42 |
| HbA1c (%) | 3.78±0.70 | 3.26±0.62 |
| TG (mmol/L) | 0.97±0.53 | 0.68±0.42 |
| TC (mmol/L) | 3.30±0.98 | 3.52±0.86 |
| HDL (mmol/L) | 1.23±0.40 | 1.42±0.38 |
| LDL (mmol/L) | 1.36±0.44 | 1.47±0.57 |

[*]$p<0.05$, [**]$p<0.01$.

FPG: fasting plasma glucose (normal range:≤6.1 mmol/L); FPI: fasting plasma insulin (normal range:≤12 μU/mL); HOMA-IR: homeostasis model assessment- insulin resistance (normal range: ≤2.67); BMI: body mass index; HbA1c: glycosylated hemoglobin A1c (normal range:<6.5%); TG: triglycerides (normal range: 0.95±0.47 mmol/L); TC: total cholesterol (normal range: 3.06±0.98 mmol/L); HDL: high-density lipoprotein cholesterol (normal range: 1.62±0.46 mmol/L); LDL: low-density lipoprotein cholesterol (normal range: 2.47±0.98 mmol/L). (**Matsuda and DeFronzo, 1999**; **Lorenzo et al., 2012**; **Cowie et al., 2010**; **Yu et al., 2019**).

important roles in the pathogenesis of T2DM. We validated the increased content of plasma PA and activation of inflammation in the T2DM macaques. In addition, we successfully induced prediabetes in mice by transplanting fecal microbiota from T2DM macaques into mice in conjunction with a diet high in PA. We also revealed the specific structure of gut microbiota that promoted T2DM development by regulating the absorption of excess PA in mice, providing experimental evidence for the functional role of gut microbiota in T2DM pathogenesis.

## Results

### Metagenome and 16S sequencing demonstrate alterations on gut microbiota in spontaneous T2DM macaques

We identified eight spontaneous T2DM macaques out of 1698 individuals from long-term glucose monitoring in a captive population (**Supplementary file 1**). The fasting plasma glucose (FPG), fasting plasma insulin (FPI), and HOMA-IR levels in the T2DM macaques were significantly higher than in the control group (p<0.01), suggesting insulin resistance in T2DM macaques. However, glycosylated hemoglobin A1c (HbA1c), triglycerides (TG), total cholesterol (TC), high-density lipoprotein cholesterol (HDL), low-density lipoprotein cholesterol (LDL), and BMI did not significantly differ from the controls (p>0.05, **Table 1**). Comparison of gut microbiota between the two groups based on 16 S rRNA amplicon found that both Shannon and Simpson indices decreased in T2DM macaques but it was not significantly different from controls (p>0.05, **Figure 1A and B**). PCoA indicated that the microbiota composition of spontaneous T2DM macaques was different from the control group (**Figure 1C**). As one of the most dominant families in both groups, Lachnospiraceae showed significantly higher abundance in the T2DM group (14.48%) than in the control group (6.66%). While the abundance of the Lactobacillaceae family was significantly greater in the control group (25.95%) compared to spontaneous T2DM macaques (20.88%) (**Figure 1D**). A total of 21 microbes were identified as differential microbes between the T2DM group and control group, where ten microbes, including Erysipelotrichaceae and five members in Lachnospiraceae family (*Ruminococcus gnavus* (current name: *Mediterraneibacter gnavus*), *Lachnospira, Coprococcus* sp. *Dorea longicatena*, and *Roseburia*) were significantly greater in the T2DM group (**Figure 1E**).

The metagenome results showed that the T2DM group had a higher abundance of Erysipelorictchaeae, *Eubacterium rectale*, Lachnospiraceae, Negativicutes, *Blautia*, and Coriobacteriia than the control group (**Figure 1—figure supplement 1A**). Functional enrichment demonstrated a total

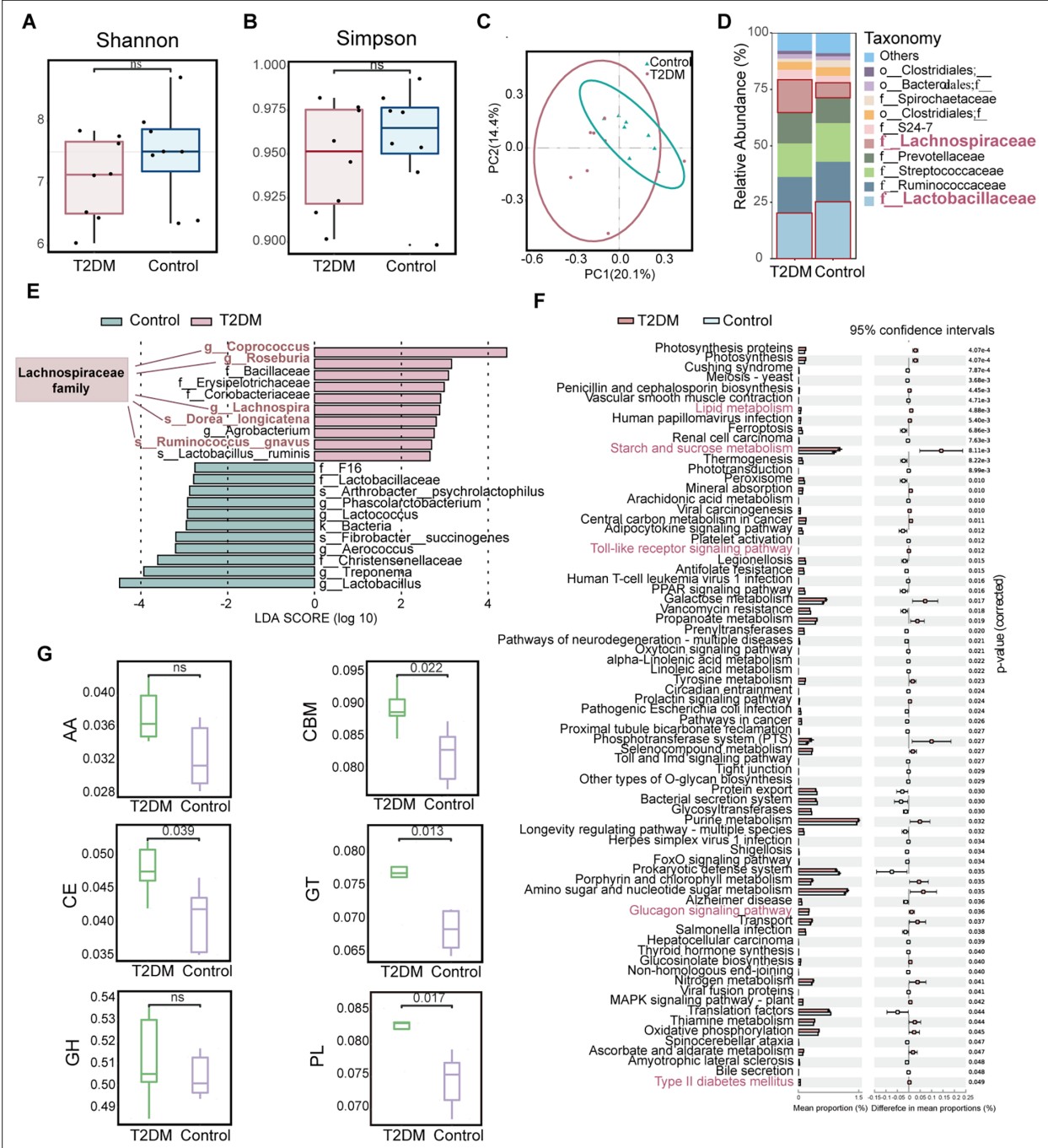

**Figure 1.** The changes in gut microbiota in spontaneous type 2 diabetes mellitus (T2DM) macaques. (**A**) Alpha diversity estimates (Shannon index) between T2DM and control groups (ns, not significant, two-tailed t-test, n=8). (**B**) Alpha diversity estimates (Simpson index) between T2DM and control groups (ns, not significant, two-tailed t-test, n=8). (**C**) Principal Coordinate Analysis (PCoA) (n=8). (**D**) Differential analysis of gut microbial composition in T2DM and control groups (n=8). (**E**) LEfSe analysis between T2DM and control groups (n=8). (**F**) Differential analysis of gut microbial function in T2DM and control groups (n=5). The pathways with red color were associated with T2DM and inflammation. Error bar is mean with ± standard deviation (s.d.). (**G**) Differential analysis of gut microbial Carbohydrate-Active enZYme (CAZy) enzyme in T2DM and control groups (n=5). CBMs: carbohydrate-binding module (p=0.022, two-tailed t-test); GTs: Glycosyl Transferases (p=0.013, two-tailed t-test); PLs: Polysaccharide Lyases (p=0.017, two-tailed t-test); AA: Auxiliary activity enzymes (ns, not significant, two-tailed t-test); GH: Glycoside hydrolases (ns, not significant, two-tailed t-test); CE: Carbohydrate esterases (p=0.039, two-tailed t-test). For all boxplots: centre lines, upper and lower bounds show median values, 25th and 75th quantiles; upper and lower whiskers show the largest and smallest non-outlier values. In c, ellipses represent the 95% confidence intervals.

The online version of this article includes the following figure supplement(s) for figure 1:

**Figure supplement 1.** Metagenome analysis of microbiota.

of 74 KEGG pathways with significant differences between the two groups. These pathways were mainly associated with T2DM and inflammation, including type II diabetes mellitus, glucagon signaling pathway, starch and sucrose metabolism, lipid metabolism, and toll-like receptor signaling pathways (*Figure 1F*). Among the six CAZy enzyme families, the Carbohydrate-binding module (CBMs), Glycosyl Transferases (GTs), Carbohydrate esterases (CEs), and Polysaccharide Lyases (PLs) families were significantly upregulated in the T2DM group, indicating significant changes in carbohydrate metabolism (*Figure 1G*). The results of metagenome and 16 S sequencing demonstrated significant alterations in the composition and function of gut microbiota in spontaneous T2DM macaques, with a greater abundance of microbes associated with T2DM and fewer beneficial microbes.

## Fecal metabolome and blood transcriptome reveals dysfunction of fatty acid β oxidation and tryptophan metabolism in T2DM macaques

The UHPLC-MS-based metabolome analysis on fecal samples of T2DM and control macaques identified 1564 metabolites belonging to various types of secondary metabolites, with lipids and lipid-like molecules being most abundant (31.1%) (*Figure 1—figure supplement 1B*). We found 64 significantly differential metabolites between the two groups using a combined multidimensional statistical analysis (OPLS-DA) and univariate statistical analysis (t-test) (VIP >1, p<0.05) (*Figure 2A and B*, *Supplementary file 2*). Among them, muscon, indole-3-acetaldehyde, and serotonin were significantly lower in the T2DM group (*Figure 2C*) and are associated with anti-inflammatory activity (*Du et al., 2018*; *Alexeev et al., 2018*; *Manzella et al., 2018*). Notably, 64 differential metabolites were significantly enriched in one pathway: tryptophan metabolism (*Figure 2D*), and we identified two significantly different metabolites, indole-3-acetaldehyde and serotonin in this pathway. Both metabolites are microbiota-derived tryptophan metabolites and the ligands for AhR (*Alexeev et al., 2018*; *Manzella et al., 2018*). The lower concentration of AhR ligands may lead to the development of inflammation and metabolic syndromes in humans (*Natividad et al., 2018*). In addition, we found the contents of many acylcarnitine metabolites were significantly higher in the T2DM group (VIP >1, p<0.1), including l-propionylcarnitine, hexanoyl-l-carnitine, (r)-butyrylcarnitine, and isovaleryl-l-carnitine (*Supplementary file 2*). Among them, l-propionylcarnitine, a kind of acylcarnitines, was the most upregulated metabolite in the T2DM macaques (FC: 16.19) (*Figure 2C*). Acylcarnitines are products of incomplete oxidation of LCFAs, which can activate pro-inflammatory signaling pathways and ultimately inhibit insulin activity (*Rutkowsky et al., 2014*). Our results indicated incomplete LCFAs β oxidation in spontaneous T2DM macaques, while similar characteristics were also observed in insulin-resistant and T2DM humans (*Tilg and Moschen, 2008*; *Koves et al., 2008*).

Blood transcriptome analysis was consistent with metabolome results, indicating dysfunction of fatty acid β-oxidation and inflammation in T2DM macaques. Gene expression in T2DM macaques exhibited significant differences from the controls (*Figure 2E*), and a total of 161 differentially expressed genes (DEGs) (26 upregulated and 135 downregulated) were identified in T2DM macaques at a FDR level of 0.05 (*Figure 2F*). Enrichment analysis of the DEGs was linked to diabetes, fatty acid metabolism, and inflammation, such as diabetic cardiomyopathy, glucose homeostasis, fatty acid metabolic process, and chemokine production (*Figure 2G*). We also identified 26 differential enrichment pathways between the two groups by aggregate fold change (AFC), and most were associated with insulin resistance and inflammation, including insulin resistance, PI3K-Akt signaling pathway, bacterial invasion of epithelial cells, and NOD-like receptor signaling pathway (*Figure 2—figure supplement 1A*). As shown in the insulin resistance pathway, expression of *IL6* and *IRS1* were upregulated, while *INSR* was downregulated in T2DM macaques (*Figure 2—figure supplement 1B*). The WGCNA analysis identified four modules that were related to T2DM. Dark green module and brown module were significantly positively correlated with T2DM, while green module and darkred module were negatively correlated with T2DM (*Figure 2H*). Genes in the dark green module and brown module related to lipolysis and inflammation were significantly upregulated in T2DM macaques, and genes in the green module and darkred module related to fatty acid metabolism and insulin secretion were significantly downregulated (*Figure 2—figure supplement 1C*). With the cut-off (|kME>0.8|), a total of 546 genes in the four modules were identified as hub genes (*Supplementary file 3*). Of these 546, 102 hub genes were also DEGs (*Figure 2I*). Several of these genes have been reported as correlated with T2DM in humans, including *IGF2BP2*, *LEPR*, *RAP1A*, *SESTRIN 3*, and *ITLN1* (*Lee et al., 2008*; *Su et al., 2016*; *Zhao et al., 2014*; *Nascimento et al., 2013*; *Tabassum et al., 2012*). In addition, *ACSM3*,

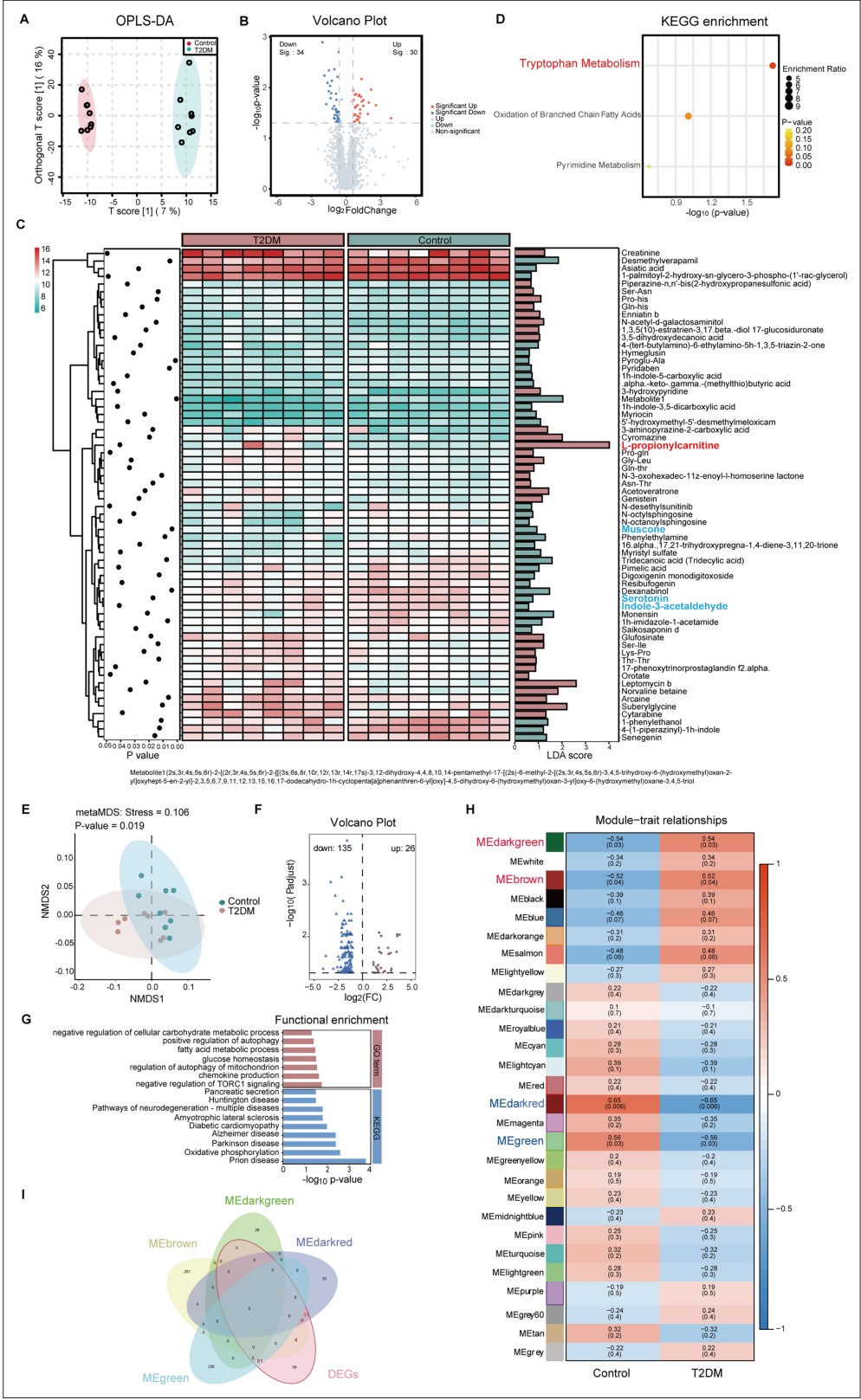

**Figure 2.** The alterations of fecal metabolites and gene expression in spontaneous type 2 diabetes mellitus (T2DM) macaques. (**A**) Orthogonal partial least squares discriminant analysis (OPLS-DA) score plots based on the metabolic profiles. (**B**) Volcano plots of metabolomics (p<0.05, two-tailed t-test). (**C**) Fecal metabolites with significant differences between T2DM and control groups (VIP >1, p<0.05, two-tailed t-test). (**D**) Enrichment

*Figure 2 continued on next page*

*Figure 2 continued*

analysis of the differentially abundant pathways between T2DM and control groups (p<0.05, two-tailed t-test). (**E**) Non-metric multidimensional scaling (NMDS) analysis between T2DM and control groups (p=0.019, two-tailed t-test). (**F**) Volcano plots of DEGs (log fold change ≥1, p<0.05, two-tailed t-test). (**G**) The GO and KEGG pathway enrichment analyses (p<0.05, two-tailed t-test). (**H**) Weighted Gene Co-Expression Network Analysis (WGCNA). (**I**) Venn analysis between hub genes and DEGs. In A and E, ellipses represent the 95% confidence intervals. Data shown are from eight individuals per group.

The online version of this article includes the following figure supplement(s) for figure 2:

**Figure supplement 1.** Pathway enrichment analysis by aggregate fold change (AFC) and Weighted Gene Co-Expression Network Analysis (WGCNA).

---

*HADHB*, and *EFHB* are involved in fatty acid β-oxidation and inflammation (*Junková et al., 2021*; *Sekine et al., 2022*; *Howson et al., 2009*).

## Validation of LCFAs accumulation and inflammation in T2DM macaques

Collectively, metabolome and transcriptome results indicated dysfunction in fatty acid β oxidation and tryptophan metabolism in T2DM macaques, which may lead to LCFA accumulation and inflammation. To support this conclusion, we performed targeted medium- and long-chain fatty acid mass spectrometry of plasma and examined serum inflammatory cytokines in the macaques. A total of 34 fatty acids were detected, and among the five types of fatty acids, the concentration of saturated fatty acid (SFA) was significantly greater in the T2DM group (p<0.05, *Figure 3A*), while other types of fatty acids were not significantly different between the two groups (p>0.05, *Figure 3B–E*). In particular, concentrations of PA, palmitoleic acid, and oleic acid were significantly higher in the T2DM group than control group (p<0.05 and VIP >1). The concentration of PA in the plasma of T2DM macaques increased, while the concentration of palmitic acid in the stool decreased (*Figure 3F and G*, *Supplementary file 2*). Increased content of PA, palmitoleic acid, and oleic acid in the plasma was also found in human T2DM (*Xu et al., 2020*; *Liu et al., 2010*). PA was an important metabolite mediating insulin resistance through three main mechanisms, being increased diacylglycerol and ceramide synthesis, mitochondrial and endoplasmic reticulum stress, and activation of pro-inflammatory pathways through membrane receptors (*Ertunc and Hotamisligil, 2016*; *Salvadó et al., 2015*; *Tumova et al., 2016*; *Cani et al., 2007*). Analysis on the serum inflammatory cytokines found that IL-1β was significantly higher in the T2DM group (p<0.05, *Figure 3H*), but TNF-α and IL-6 levels showed no significant difference between the two groups (p>0.05, *Figure 3I and J*). IL-1β is a major player in a variety of autoinflammatory diseases and a key promoter of T2DM systemic and tissue inflammation (*Dinarello et al., 2010*). Moreover, blood routine examination showed an increase in white blood cell (WBC) number, neutrophil (NEU) percentage and NEU number and a decrease of lymphocyte (LYM) percentage and LYM number, also indicating the inflammation in the T2DM macaques (p<0.05, *Table 2*).

To investigate the effect of gut microbiota on the LCFAs accumulation and inflammation in T2DM macaques, we performed a correlation analysis among the DEGs, differential metabolites, and differential microbes using Spearman's rank correlation. Four differential microbes in Lachnospiraceae family (*Coprococcus*, *Lachnospira*, *Roseburia*, and *Dorea longicatena*) were significantly associated with three differential metabolites of PA, palmitoleic acid, and oleic acid (|r|>0.5, adj p<0.05), suggesting the participation of Lachnospiraceae microbes in LCFAs accumulation in T2DM macaques (*Figure 3K*). In addition, the bacteria in class Coriobacteriia was also associated with the three LCFAs (|r|>0.5, adj p<0.05, *Figure 1—figure supplement 1C and D*).

## Fecal microbiota transplantation (FMT) with high content PA food induce prediabetes in mice

To determine the causative effect of gut microbiota and PA on T2DM development, we collected feces from the spontaneous T2DM macaques and performed FMT in antibiotic-pretreated mice. Mice were either administered by FMT (FT), fed with a high concentration PA diet (PA), or were combined FMT and PA diet (FTPA). A control group was used and they were fed with normal commercial food and lacked FMT (*Figure 4A*). FPG monitoring found that FPG levels in the FTPA group and FT group increased continuously from day 60 (*Figure 4B*), while the control group maintained stable FPG levels throughout the 120 days. The FTPA group showed the highest FPG of 6.7 mmol/L at day 120, which

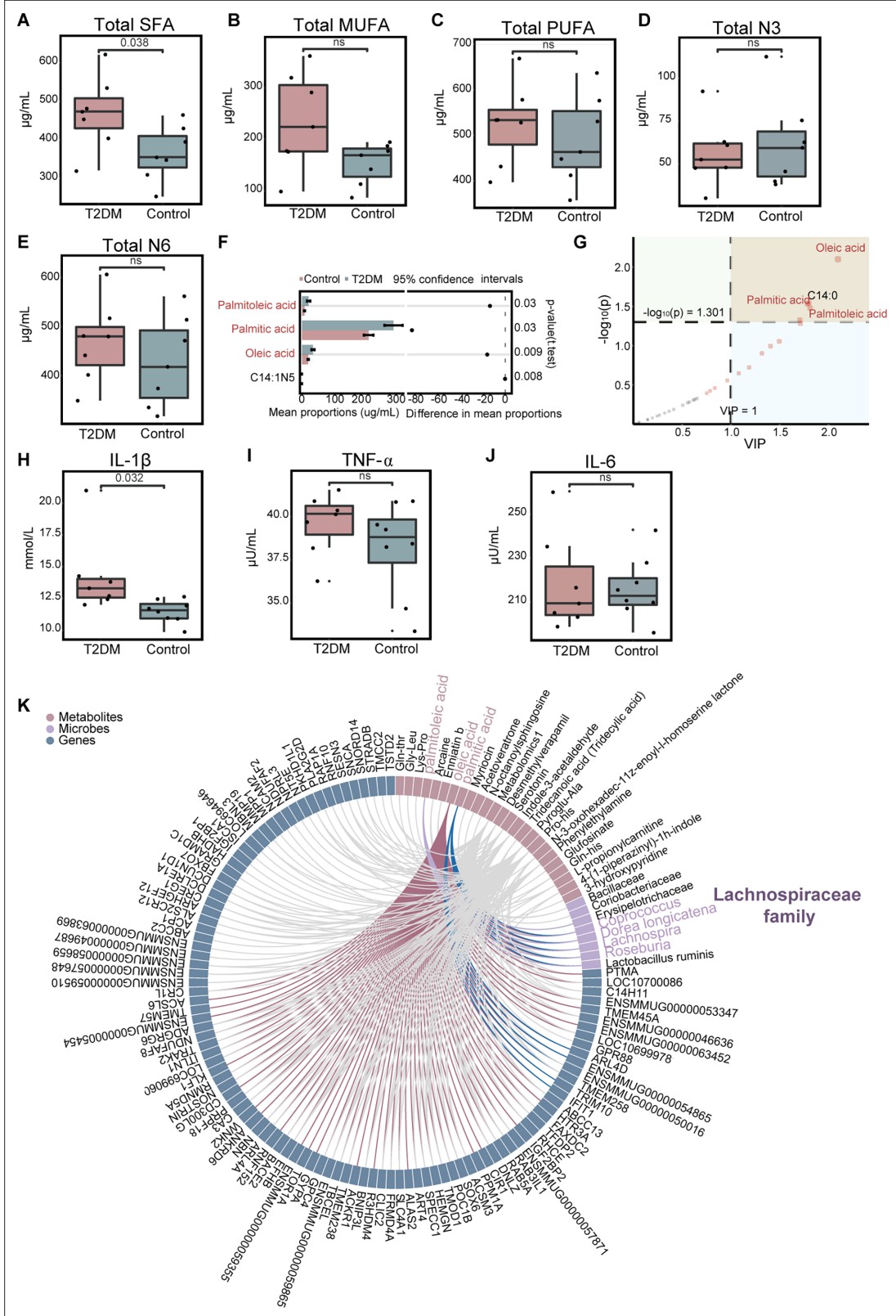

**Figure 3.** Long-chain fatty acids (LCFAs) accumulation and inflammation occurred in spontaneous type 2 diabetes mellitus (T2DM) macaques. (A–E) The contents of saturated fatty acid (SFA) (A, p=0.038), MUFA (B), PUFA (C), N3 (D), and N6 (E) in plasma (ns, not significant, two-tailed t-test). (F) The univariate analysis by two-tailed t-test, error bar is mean with ±s.d. (G) The multidimensional analysis by VIP value (VIP >1). (H–J) The contents of serum inflammatory cytokines, including IL-1β (H, p=0.032), TNF-α (I), and IL-6 (J) (ns, not significant, two-tailed t-test). (K) Correlation analysis between DEGs,

*Figure 3 continued on next page*

*Figure 3 continued*

differential metabolites, and differential microbes using Spearman rank correlation (|r|>0.5, adj p<0.05). For all boxplots: centre lines, upper and lower bounds show median values, 25th and 75th quantiles; upper and lower whiskers show the largest and smallest non-outlier values. Data shown are from seven to eight individuals per group.

was significantly higher than the control group and indicated FTPA-treated mice had developed prediabetes (*Figure 4C*). The oral glucose tolerance test (OGTT) showed obvious glucose intolerance in the FTPA group (*Figure 4D and E*), and the FPI values and insulin tolerance test (ITT) were both significantly elevated in the FTPA group, indicating insulin resistance (*Figure 4F and G*). Meanwhile, body weight (BW), TC, and TG significantly increased in the FTPA group (*Figure 4H–J*). While mice in the FT group and PA group had significant weight gain and mild insulin resistance, they did not exhibit significant glucose intolerance and no significant elevation in TC and TG levels compared to

**Table 2.** Blood routine examination of Control and type 2 diabetes mellitus (T2DM) groups.

| Index | T2DM (n=7) | Control (n=7) |
| --- | --- | --- |
| WBC (10e9/L) | 15.63±4.66 | 11.32±2.19[*] |
| RBC (10e12/L) | 5.47±0.51 | 5.77±0.43 |
| HGB (g/L) | 129.57±15.48 | 136.71±9.60 |
| HCT (%) | 41.31±3.64 | 44.22±3.12 |
| MCV (fl) | 74.13±2.49 | 76.74±2.00 |
| MCH (pg) | 23.17±1.00 | 23.71±0.60 |
| MCHC (g/L) | 312.86±10.16 | 309±8.14 |
| RDW (%) | 15.23±2.12 | 14.69±1.74 |
| PLT (10e9/L) | 402±86.66 | 371.57±86.42 |
| MPV (fl) | 10.54±1.62 | 10.24±1.12 |
| PCT (%) | 0.42±0.08 | 0.38±0.09 |
| PDW (%) | 14.94±0.58 | 14.83±1.64 |
| LYM% (%) | 23.67±10.26 | 47.71±8.13** |
| LYM# (10e9/L) | 3.46±1.66 | 5.39±1.52[*] |
| MON% (%) | 4.18±3.34 | 6.18±2.29 |
| MON# (10e9/L) | 0.65±0.62 | 0.71±0.38 |
| NEU% (%) | 71.13±13.23 | 44.28±8.96** |
| NEU# (10e9/L) | 11.36±4.91 | 5.00±1.38** |
| EOS% (%) | 0.91±0.69 | 1.50±1.94 |
| EOS# (10e9/L) | 0.14±0.12 | 0.17±0.22 |
| BAS% (%) | 0.12±0.13 | 0.33±0.47 |
| BAS# (10e9/L) | 0.02±0.02 | 0.04±0.07 |
| NRBC# (10e9/L) | 0 | 0 |
| NRBC% (%) | 0 | 0 |

*p<0.05, **p<0.01.

WBC: white blood cell; RBC: red blood cell; HGB: hemoglobin; HCT: hematocrit; MCV: mean corpuscular volume; MCH: mean corpuscular hemoglobin; MCHC: mean corpuscular hemoglobin concentration; RDW: red blood cell distribution width; PLT: platelet; MPV: mean platelet volume; PCT: procalcitonin; PDW: platelet volume distribution width; LYM%: lymphocyte percentage; LYM#: lymphocyte value; MON%: monocytes percentage; MON#: monocytes value; NEU%: neutrophil percentage; NEU#: neutrophil value; EOS%: eoseosinophil percentage; EOS#: eoseosinophil value; BAS%: basophil percentage; BAS#: basophil value; NRBC%: nucleated red blood cell percentage; NRBC#: nucleated red blood cell value.

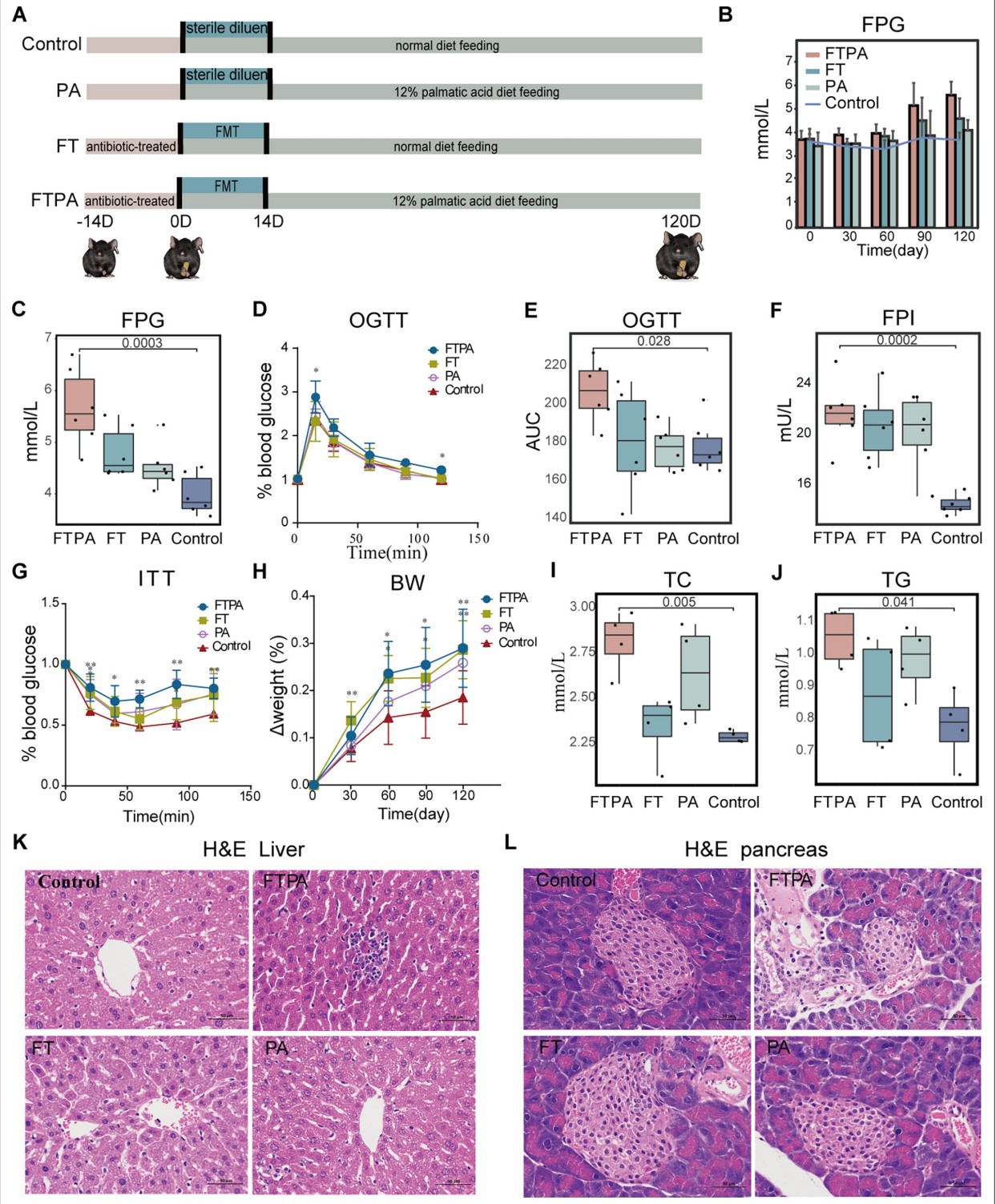

**Figure 4.** The fecal microbiota transplanting (FMT) and high palmitic acid (PA) diet mice developed pre-type 2 diabetes mellitus (T2DM) characteristics. (**A**) Experimental scheme of FMT and high PA diet treatment. (**B–H**) Metabolic analysis, including the trend of fasting plasma glucose (FPG) within 120 days (**B**), FPG (C, p=0.0003), oral glucose tolerance test (OGTT) (**D**), AUC of OGTT (E, p=0.028), fasting plasma insulin FPI (F, p=0.007), ITT (**G**), and body weight change (**H**) on day 120. (**I–J**) The contents of TC (I, p=0.005) and TG (J, p=0.041) in serum on day 120. (**K, L**) Representative H-E staining images of liver (**K**) and pancreas (**L**). For all boxplots: centre lines, upper and lower bounds show median values, 25th and 75th quantiles; upper and lower whiskers show the largest and smallest non-outlier values. Significance was determined using one-way ANOVA. In d, g, and h: *p<0.05, **p<0.01. Data shown are from four to six individuals per group.

controls. Histopathological changes in the pancreas and liver of mice were investigated using H-E staining. Hepatocytes focal necrosis with inflammatory cell infiltration was commonly observed in the FTPA mice, but not frequent in hepatocytes in FT and PA groups (*Figure 4K*). Furthermore, decreased volume and area in pancreatic islets and inflammatory cell infiltration were detected in the FTPA mice (*Figure 4L*), while such pathological changes were not found in the control group. To eliminate interference from host species divergence in gut microbiota composition, we supplemented mouse experiments using FMT from control macaques (HFT group) (*Figure 5—figure supplement 2A*). By day 30, the HFT group exhibited significantly lower body weight than the untreated control group (p<0.05) (*Figure 5—figure supplement 2B*). Throughout the experimental period, FPG levels in both HFT and control groups remained within the normal range (<6 mmol/L) without significant differences, indicating that transplantation of control macaque microbiota did not induce glycemic alterations (*Figure 5—figure supplement 2C*).

## Specific structure of gut microbiota mediates the absorption of excess PA in the ileum

Given that the content of plasma PA significantly increased in the T2DM macaques (*Figure 3F and G*), we examined PA content in the feces, ileum, and serum in mice to compare the prediabetes mice and the spontaneous T2DM macaques. PA content in the serum and ileum of the FTPA group was significantly higher than the control group (p<0.05, *Figure 5A and B*), but its content in feces was significantly lower than the control group (p<0.05, *Figure 5C*). This suggested that the absorption of PA was significantly enhanced in the ileum leading to the increase of PA in serum. To verify this conclusion, expression of the *Cd36* gene, a gene involved in the uptake and oxidation of LCFAs (*Sun et al., 2023*), was examined in the ileum of mice. Ileums showed a significantly upregulated expression of *Cd36* in the FTPA group compared to the control group (p<0.05, *Figure 5D*). In contrast, the level of IL-17A, a protein inhibiting the expression of *Cd36* (*Kawano et al., 2022*), was significantly reduced in the ileum of the FTPA group (p<0.05, *Figure 5E*). Interestingly, the PA group's content of PA, *Cd36* expression, and IL-17A was not significantly different from the control group. This indicated that without FMT, the ileum could not absorbp PA effectively even when fed in high concentrations. Consequently, gut microbiota mediated the absorption of excess PA in the ileum.

We then compared diversity and composition of microbiota communities between the FTPA, FT, PA, and control groups, and a fifth group consisting of the microbiota transplants (TP) from T2DM macaques. Shannon and Simpson indices were lower in the FTPA, FT, and PA groups than the control group (*Figure 5—figure supplement 1A and B*). NMDS analysis also indicated a distinct microbiota composition from the control group compared to other groups (NMDS1). In addition, microbiota composition of TP from T2DM macaques was distinct from other groups but a little closer to the FTPA and FT groups (NMDS2, *Figure 5F*). In particular, the Lachnospiraceae family showed the higher abundance in the TP, FTPA, and FT groups than in the PA and control groups (*Figure 5—figure supplement 1C*). The abundances of three members of microbes in Lachnospiraceae (*R. gnavus* (current name: *M. gnavus*), *Coprococcus* sp., and *Clostridium*) in the FTPA and FT groups gradually increased over time after FMT (*Figure 5—figure supplement 1D*). The FTPA and FT groups shared many differential microbes compared to the control group, such as the significantly upregulated *R. gnavus* (current name: *M. gnavus*) and *Coprococcus* sp., and the significantly downregulated Christensenellaceae, F16, *Treponema* sp., and *Fibrobacter succinogenes* (*Figure 5G and H*). It is noteworthy that these microbes were also differential microbes between T2DM macaques and controls, and their abundances changed in the same way as in macaques (*Figure 1E*). However, mice in the PA group did not share differential microbes with the spontaneous T2DM macaques. Correspondingly, the change of serum PA content in the PA group was not significantly higher than the control group (*Figure 5—figure supplement 1E*). Integrating 16 S rRNA sequencing data from the HFT, FT, and FTPA groups showed that the antibiotic treatment effectively depleted the gut microbiota, resulting in microbial diversity decreasing sharply, with the dominant phyla shifting from Bacteroidota and Bacillota to Pseudomonadota (*Figure 5—figure supplement 2D–G*). The HFT group restored microbial diversity within 30 days, achieving community proportions comparable to untreated controls. Core functional phyla (Bacteroidota and Bacillota) stably colonized in HFT group (Figure S4D-I). Critically, FT and FTPA groups exhibited increased Lachnospiraceae (including genera *Ruminococcus* (current name: *Mediterraneibacter*), *Coprococcus*, and *Clostridium*) compared with the HFT group on day 30.

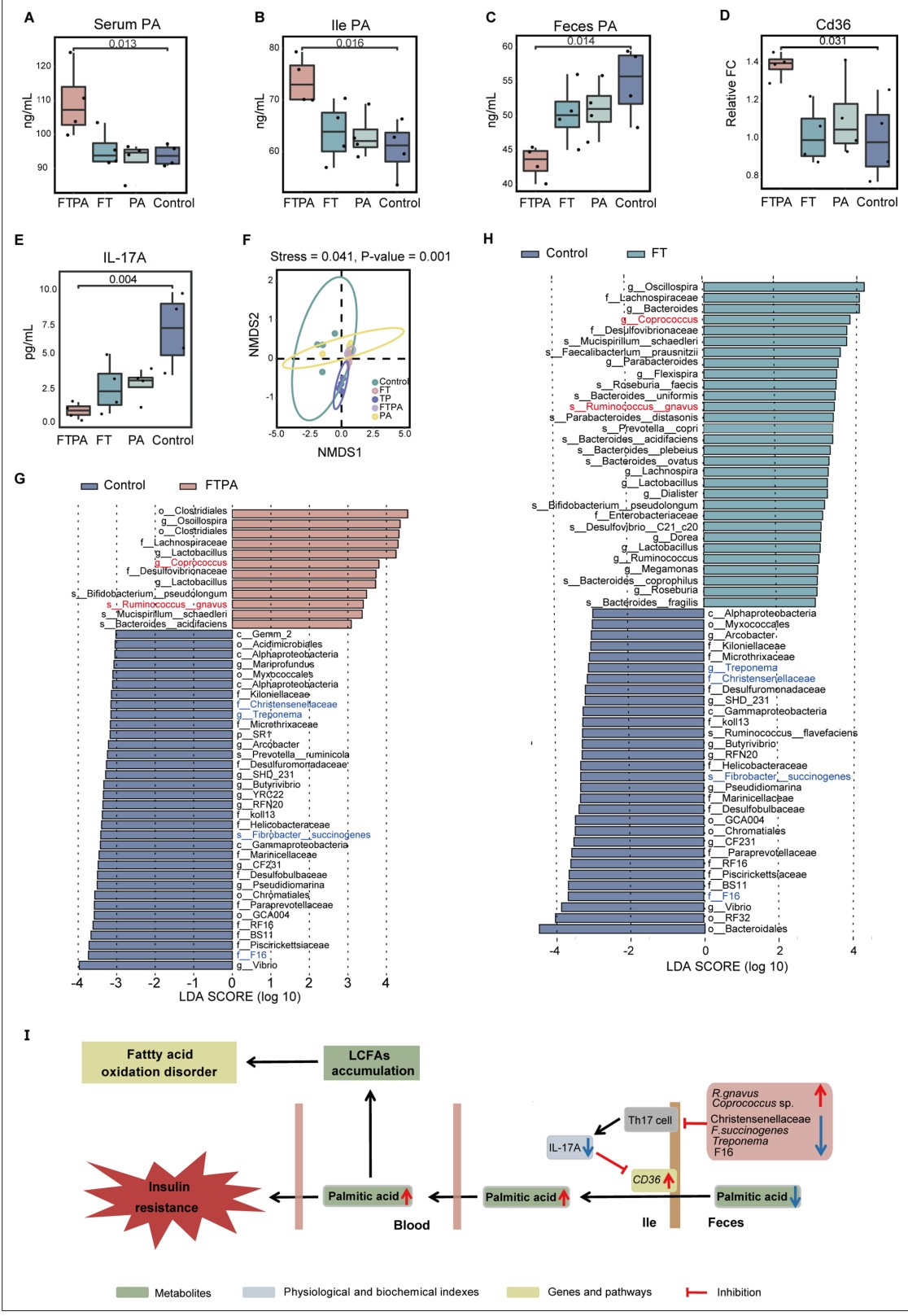

**Figure 5.** The palmitic acid (PA) accumulation required the specific gut microbiota. (**A–C**) Total PA contents in serum (A, p=0.013), ileum (B, p=0.016), and feces (C, p=0.014) on day 120. (**D**) Quantitative RT-PCR for Cd36 transcripts in ileum on day 120 (p=0.049). (**E**) The content of IL-17A in ileum on day 120 (p=0.027). For all boxplots: centre lines, upper and lower bounds show median values, 25th and 75th quantiles; upper and lower whiskers show the largest and smallest non-outlier values. Significance was determined using one-way ANOVA. Data shown are from three to four individual macaques

*Figure 5 continued on next page*

*Figure 5 continued*

per group. (**F**) Non-metric multidimensional scaling (NMDS) analysis (p=0.001, one-way ANOVA), ellipses represent the 95% confidence intervals. (**G, H**) LEfSe analysis between FTPA and control groups (**G**), FT and control groups (**H**). Data shown are from four individuals per group. (**I**) Specific gut microbiota structure promoted the absorption of excess PA by regulating the expression of IL-17A and *Cd36*, leading to the LCFAs accumulation and insulin resistance.

The online version of this article includes the following figure supplement(s) for figure 5:

**Figure supplement 1.** The changes in gut microbiota in FTPA, FT, and palmitic acid (PA)-treated mice.

**Figure supplement 2.** The changes in gut microbiota in HFT-treated mice.

In addition, LEfSe comparison identified significant *R. gnavus* (current name: *M. gnavus*) enrichment in the FT group (LDA >3, p<0.01) (***Figure 5—figure supplement 2J–M***). Our results suggested that the transplanted microbiota from spontaneous T2DM macaques, especially the increased abundance of *R. gnavus* (current name: *M. gnavus*) and *Coprococcus* sp. and decreased abundance of *Treponema*, *F. succinogenes*, Christensenellaceae and F16, promoted the absorption of excess PA by regulating the expression of IL-17A and Cd36, leading to the LCFAs accumulation and insulin resistance (***Figure 5I***).

## Discussion

With a multi-omics technology, this study comprehensively characterizes the gut microbiota, metabolites and gene expression of spontaneous T2DM macaques. The gut microbiota diversity in T2DM macaques decreased. In particular, the abundance of bacteria *R. gnavus* (current name: *M. gnavus*) and Erysipelotrichaceae were upregulated while the abundance of Christensenellaceae was downregulated in the T2DM macaques. Metabolome results demonstrated a decrease of microbiota-derived tryptophan and anti-inflammatory metabolites, indicating that T2DM macaques were prone to inflammation. Notably, the accumulation of acylcarnitine metabolites, the suggested biomarkers for human T2DM (***Liu et al., 2010***), indicated incomplete mitochondrial LCFA β-oxidation in T2DM macaques. Transcriptome results identified many DEGs linked to insulin resistance, fatty acid β oxidation, and inflammation, including *IGF2BP2*, *LEPR*, *RAP1A*, *SESTRIN 3*, and *ITLN1* that have also been reported in human T2DM (***Lee et al., 2008***; ***Su et al., 2016***; ***Zhao et al., 2014***; ***Nascimento et al., 2013***; ***Tabassum et al., 2012***). Combining the multi-omics results, we revealed the complex pathological mechanisms in the spontaneous T2DM macaques (***Figure 6***), which is comparable to T2DM humans. Firstly, expression of genes related to lipolysis, fatty acid oxidation, LCFA accumulation, inflammation, and insulin secretion are dysregulated, promoting the development of T2DM (***Figure 6A***). This is particularly related to lipid metabolism, where the increase of lipolysis and the downregulated expression of fatty acid metabolism-related genes *HADHB* and *ACSM3* cause the accumulation of acylcarnitine and LCFAs, which results in incomplete LCFA oxidation in T2DM macaques (***Figure 6B***). We also suggest that the decrease of *Lactobacillus* sp. causes the reduction of Aryl hydrocarbon receptor (AhR) ligands (serotonin and indole-3-acetaldehyde) given that *Lactobacillus* is producer of AhR ligands (***Hezaveh et al., 2022***). This then promotes the expansion of the PA producer Erysipelotrichacea (***Brawner et al., 2019***; ***Turpin et al., 2020***) and ultimately leads to the increase of PA levels. The increase in the abundance of Erysipelotrichacea and *R. gnavus* (current name: *M. gnavus*) also promotes the development of T2DM by activation of inflammation (***Turnbaugh et al., 2008***; ***Zhang et al., 2009***; ***López-Almela et al., 2021***; ***Figure 6C***). The accumulation of PA was reported to lead to insulin resistance by affecting genes in the insulin signaling pathway (***Palomer et al., 2018***). Our study demonstrates that the significantly changed expression of *RAP1A*, *SESTRIN3*, and *IRS1* in PA-mTORC1-Akt pathway causes insulin resistance in T2DM macaques. Moreover, the increase in PA can promote the development of T2DM by upregulating the NF-κB signaling pathway (***Figure 6D***).

We validated the observations from the multi-omics analysis, finding significantly higher inflammatory cytokines IL-1β and LCFA accumulation, especially significant PA accumulation in the T2DM macaques. And found LCFA metabolites were significantly correlated with bacteria in the Lachnospiraceae family. Numerous studies imply an association of gut microbiota with T2DM development. By transplanting gut microbiota from healthy individuals to T2DM individuals, symptoms such as insulin resistance or inflammation were improved (***Zhao et al., 2018***; ***Kootte et al., 2017***). However, there is no evidence to date suggesting gut microbiota have directly causative effects on T2DM development. Our study confirmed the causative effect of gut microbiota and PA on T2DM development by

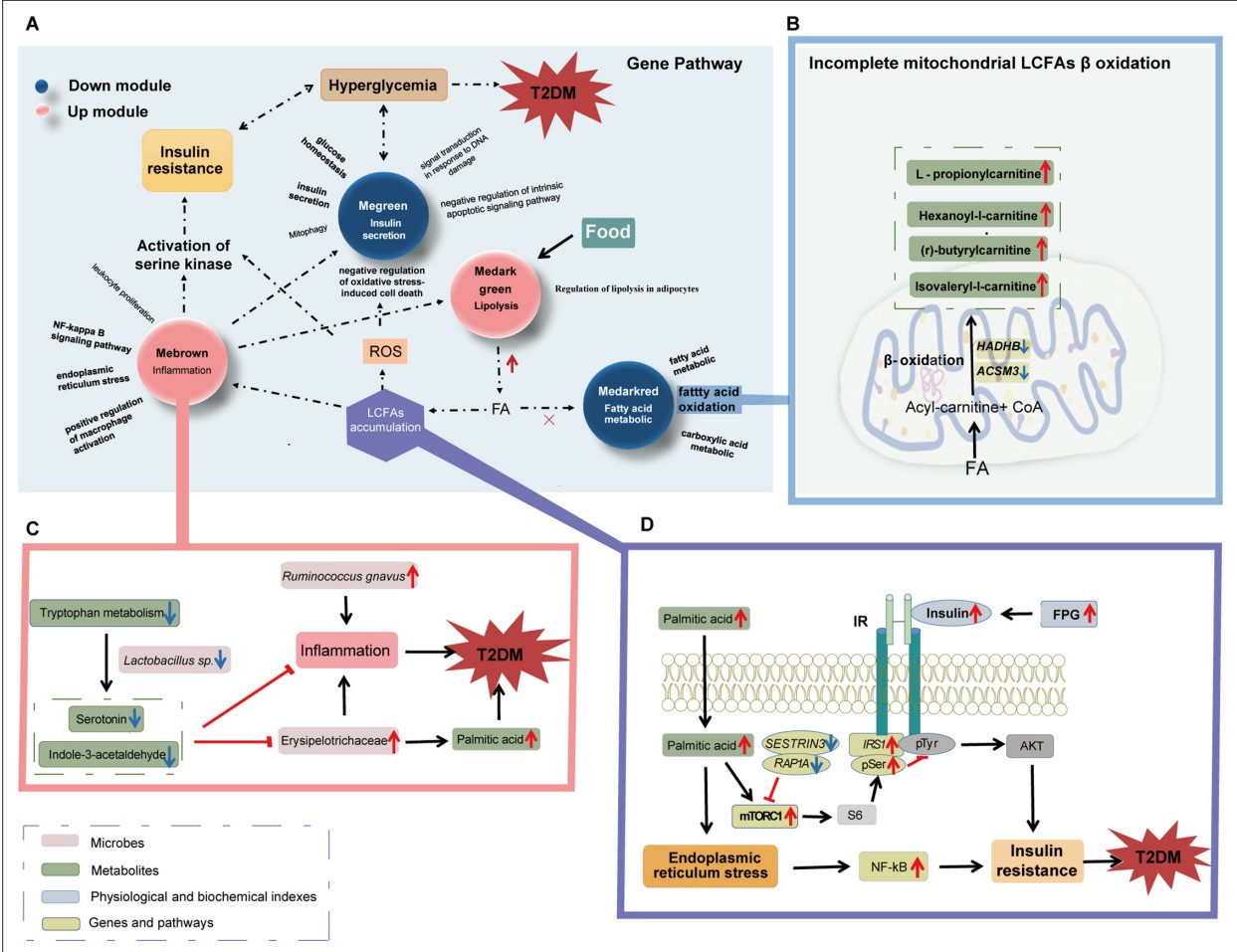

**Figure 6.** Integration of multi-omics results. (**A**) Insulin resistance, fatty acid oxidation disorders, long-chain fatty acid s (LCFAs) accumulation, and inflammation occurred in spontaneous type 2 diabetes mellitus (T2DM) macaques. (**B**) Incomplete mitochondrial LCFAs β oxidation. The expression levels of fatty acid metabolism-related genes HADHB and ACSM3 were downregulated in spontaneous T2DM macaques, which could lead to accumulation of acylcarnitine, including l-propionylcarnitine, hexanoyl-l-carnitine, (**r**)-butyrylcarnitine, and isovaleryl-l-carnitine. (**C**) Gut inflammation. The decrease of *Lactobacillus sp.* likely caused the reduction of serotonin and indole-3-acetaldehyde, which promotes the expansion of palmitic acid (PA) producer Erysipelotrichacea and ultimately led to PA accumulation. Both Erysipelotrichacea and *Ruminococcus gnavus* (current name: *Mediterraneibacter gnavus*) promote the development of inflammation. Accumulation of PA and inflammation are important factors in the development of T2DM. (**D**) Accumulation of PA promoted the development of insulin resistance. In the PA-mTORC1-Akt pathway, the changes of *RAP1A*, *SESTRIN3*, and *IRS1* expression promoted the development of insulin resistance in spontaneous T2DM macaques. The increase in PA promoted the development of T2DM by up-regulating the NF-κB signaling pathway.

transplanting fecal microbiota from spontaneous T2DM macaques to antibiotic- pretreated mice. We successfully induced prediabetes in mice after combining FMT administration and high PA ingestion. However, when the treatments were administered on their own, the mice did not develop prediabetes. We determined, for the first time, that gut microbiota mediated the absorption of excess PA in the ileum by quantitatively examining PA contents in feces, ileum, and serum, and analysis of the expression of *Cd36* and IL-17A level in the ileum. Most notably, this then resulted in accumulation of PA in the serum and finally led to T2DM development. Without the transplanting gut microbiota, the ileum could not absorb the PA effectively even at a high concentration of ingested PA. Our study highlights the essential roles of gut microbiota in T2DM development, which may account for the inability of prior studies to induce T2DM in macaques through high-fat diet intervention alone (*Ji et al., 2012*; *Sun et al., 2020*). Furthermore, applying this approach to induce T2DM in macaques will enable deeper investigation into gut-microbiota-driven mechanisms underlying disease pathogenesis.

We then determined the specific gut microbiota structure that related to T2DM development in the prediabetes mice and spontaneous T2DM macaques. We found that the increased abundance of

**Table 3.** Primers of RT-PCR.

| Gene name | Sequences of primers |
| --- | --- |
| CD36/F | 5'-ATGGGCTGTGATCGGAACTG-3' |
| CD36/R | 5'-GTCTTCCCAATAAGCATGTCTCC-3' |
| GADPH/F | 5'-CCTCGTCCCGTAGACAAAATG-3' |
| GADPH/R | 5'-TCTCCACTTTGCCACTGCAA-3' |

R. gnavus (current name: M. gnavus) and Coprococcus sp., and the decreased abundance of Treponema, F. succinogenes, Christensenellaceae, and F16, were involved in the T2DM development. R. gnavus (current name: M. gnavus) can promote insulin resistance by regulating the content of tryptamine/phenethylamine (**Zhai et al., 2023**). Moreover, R. gnavus (current name: M. gnavus) is a mucin-degrading microbe that leads to an increase in inflammation (**Paone and Cani, 2020**; **Png et al., 2010**; **Henke et al., 2019**). The intestinal mucous layer is an important barrier separating intestinal tissue from microbiota, and microbiota composition plays a major role in affecting the integrity of intestinal mucous layer (**Paone and Cani, 2020**). The increase of R. gnavus (current name: M. gnavus) suggested a higher risk of damage to the integrity of the intestinal mucous layer. This is further supported by our results of the lower abundance of Lactobacillus sp., which are AhR ligand producers, and the decreased content of AhR ligands (tryptophan microbial metabolites) in spontaneous T2DM macaques. The deficiency of AhR ligand reduced the production of intestinal mucus and increased the risk of microbial invasion, which in turn affected the immune cell differentiation and cytokine production (**Yin et al., 2019**; **Ma et al., 2018**). The cytokine IL-17A is a regulator of the fatty acid transporter Cd36 and lipid absorption can be promoted by reducing the inhibition of Cd36 expression by IL-17A (**Zhang et al., 2009**). The decreased IL-17A level and increased Cd36 expression in prediabetes mice indicated that the specific gut microbiota promoted the absorption of excess PA by disrupting the integrity of the intestinal mucous layer and regulating the expression of IL-17A and Cd36. In addition, the beneficial bacteria decreased in abundance in T2DM macaques and prediabetes mice. These beneficial bacteria, such as F. succinogenes, Christensenellaceae, and F16, protect the mucosal barrier and improved insulin resistance (**Oki et al., 2016**; **Miguel et al., 2019**; **Zhong et al., 2021**). We inferred the collectively effects of these gut microbes determined the absorption of excess PA from ileum to serum, which might contribute to the development of T2DM. Previous studies have shown that insulin-resistant patients exhibit increased fecal monosaccharides associated with microbial carbohydrate metabolism (**Takeuchi et al., 2023b**). Furthermore, commensal species of Lachnospiraceae actively overproduce long-chain fatty acids during metabolic dysfunction through altered bacterial lipid metabolism. The microbe-derived fatty acids impair intestinal epithelial integrity to exacerbate metabolic dysregulation (**Takeuchi et al., 2023a**). Given that microbial metabolic activity causally modulates host metabolic homeostasis, the content change of PA was potentially associated with a dynamic equilibrium between host absorption and microbial metabolism. Further integrative studies on the fecal fatty acid metabolome, microbial PA metabolism, and functional pathways will be crucial for delineating causal links between dysbiosis and lipid metabolic dysfunction in T2DM.

In conclusion, spontaneous T2DM macaques that have never been treated with diabetes-related drugs provide a valuable model for our understanding of the pathological characteristics and

**Table 4.** RT-PCR reaction components.

| Reagent | Volume |
| --- | --- |
| 2× SG Fast qPCR Master Mix | 10.0 µL |
| DNF Buffer | 2.0 µL |
| F primer (10 µmol/L) | 0.4 µL |
| R primer (10 µmol/L) | 0.4 µL |
| cDNA | 1.0 µL |
| ddH$_2$O | 6.2 µL |

**Table 5.** RT-PCR cycle procotol.

| Step | Temperature | Time |
| --- | --- | --- |
| 1 | 95°C | 3 min |
| 2 | 95°C | 1–3 s |
| 3 | 60°C | 30 s |
| 4 | | |
| 5 | 72°C | 1 min |

pathogenesis of T2DM. This study characterized changes in gene expression, metabolites, and gut microbiota levels of spontaneous T2DM macaques using multi-omics techniques. We found gut abnormal microbiota, tryptophan metabolism and fatty acid β-oxidation disorders, inflammation, and PA accumulation. We also successfully induced prediabetes in mice by transplanting fecal microbiota from T2DM macaques into antibiotic-pretreated mice fed a high PA diet. Our study confirms the functional role of gut microbiota and PA in the T2DM progression. The microbiota composition, specifically higher abundance of *R. gnavus* (current name: *M. gnavus*) and *Coprococcus* sp. and lower abundance of *Treponema*, *F. succinogenes*, Christensenellaceae, and F16, promoted the absorption of excess PA which is important for the development of T2DM. However, in this study, such microbial alterations were detected in macaques after they had developed the disease of T2DM, instead of before or onset of T2DM, the causative effect of gut microbiota and their action mechanism on the development of T2DM is worth further investigation. This study provides new insights into the interaction of gut microbiota and metabolites in the development of T2DM, which expands our understanding of the pathogenesis of this metabolic disease and may provide novel insights for the treatment of T2DM in the future.

## Materials and methods

### The screening of spontaneous T2DM macaques

The experimental macaques used in this study were all from Greenhouse Biotechnology Co., LTD (Meishan, China). We obtained eight spontaneous T2DM macaques with FPG ≥7 mmol/L and eight healthy control macaques with FPG ≤6.1 mmol/L (three consecutive detections, each detection interval of one month) from a population of 1698 captive macaques. None of these 16 screened macaques received any medical treatment for diabetes.

### Sample collection

Each experimental macaque was kept in a single cage and was fasted for 12 hr but had free access to drinking water. We obtained serum, plasma, and whole blood for the detection of physiological and biochemical parameters, and metabolome and transcriptome analysis. Fecal samples were collected within 10 min after deposition. During the sampling process, fecal samples were loaded into a 50 mL sterile centrifuge tube were stored at –80°C. This study followed animal welfare guidelines throughout the sample collection process, and all observations and samplings were approved by the Sichuan University's Animal Care Committee (Permit Numbers: SCU210429001 and SCU230810001).

### Physiological and biochemical parameters

In this study, FPG, HbA1c, and FPI were detected by hexokinase method, high performance liquid chromatography, and electro-chemiluminescence method, respectively. TC, TG, HDL, and LDL were detected by automatic biochemical analyzer. HOMA-IR is one of the criteria for T2DM, calculated as: (FPI×FPG)/22.5.

### Feces 16S rRNA amplicon sequencing and analysis

Total DNA from fecal samples was extracted using the QIAamp Fast DNA Stool Mini Kit. The V3-V4 region of the 16 S rRNA gene was amplified using the 341 F/806 R primer (341 F: 5'-CCTAYGG-GRBGCASCAG –3', 806 R: 5'-GGACTACNNGGGTATCTAAT –3'). The purification and quantification

of the amplified products were performed and followed by the sequencing library preparation with TruSeq Nano DNA LT Library Prep Kit (Illumina, USA). The library sequencing was performed on Illumina MiSeq platform and 250 bp paired-end reads were generated. Raw sequencing reads were merged using FLASH (Vl.2.7, https://ccb.jhu.edu/software/FLASH/) (*Magoč and Salzberg, 2011*) and analyzed by QIIME2 (version 2020.11.1) pipeline with default parameters. Reads after denoising by DADA2 were clustered into OTUs at 99% similarity threshold. Taxonomy of OTUs was assigned based on Greengenes reference database (*McDonald et al., 2012*). The QIIME2 diversity plugin was used to calculate alpha diversity (*Bolyen et al., 2019*). Principal Coordinate Analysis (PCoA) was determined by using the R package vegan. Differentially microbes were determined using linear discriminant analysis effect size (LEfSe) (*Segata et al., 2011*).

## Feces shotgun metagenome sequencing and analysis

Total DNA from each sample was extracted using Tiangen DNA Stool Mini Kit (Tiangen Biotech Co., Ltd., China). Metagenome sequencing was performed using the Illumina NovaSeq 6000 platform with a paired-end sequencing length of 150 bp. Trimmomatic was used for removing the adapters and low-quality raw reads based on a four-base wide sliding window, with average quality per base >20 and minimum length 90 bp (*Bolger et al., 2014*). The rhesus macaque potential sequences were removed using Bowtie2 (*Langmead and Salzberg, 2012*) with the reference genome (assembly Mmul_10). Taxonomy of remaining reads was assigned using Kraken2 (*Wood and Salzberg, 2014*) with the option '--use-mpa-style.' De novo assembly of remaining reads was performed using MEGAHIT (*Li et al., 2015*) with the option '-m 0.90 `--min-contig-len` 300.' Prodigal (*Hyatt et al., 2010*) was used for gene prediction. The construction of non-redundant gene catalogue was performed with CD-HIT (*Fu et al., 2012*) with the option '-c 0.95 -aS 0.90.' Quantification of the non-redundant genes in each sample was performed using Salmon (*Patro et al., 2017*). The amino acid sequences translated from non-redundant genes were aligned (`--id` 80% `--query-cover` 70% `--evalue` 1e-5) by DIAMOND (*Buchfink et al., 2015*) in the Carbohydrate-Active enZYmes (CAZy) database (*Lombard et al., 2014*). The annotation metabolic pathway was performed using HUMANn3 (*Franzosa et al., 2018*).

## RNA sequencing and DEG analysis

Total RNA was extracted using PAXgene Blood RNA kit. The cDNA Library was constructed following the NEBNext UltraTM RNA Library Prep Kit for Illumina (NEB, USA) manual, and index was added to each sample for sample differentiation. cBot Cluster Generation System was used to cluster the sequences with the same index. Illumina HiSeq 2500 sequencing platform was used to obtain the paired-end sequencing reads (PE150). NGS QC Toolkit v2.3.3 (*Patel and Jain, 2012*) was used to obtain high quality reads (high-quality paired reads with more than 90% of bases with Q-value ≥20 were retained). Processed reads of each sample were mapped to the macaque reference genome using HISAT2 v2.1.0 (*Kim et al., 2015*). Each alignment output file was assembled into a separate transcriptome using StringTie v1.3.6 (*Pertea et al., 2015*), resulting in a transcript GTF file. To obtain the expression value of TPM (transcripts per million) and raw read counts of each gene, transcript GTF file was used as the reference annotation file. Differential expression analysis was performed using DESeq2 R package (*Love et al., 2014*). DEGs were screened according to Foldchange value (FC) and p-value corrected by FDR (Benjamini-Hochberg method was selected) (adj $p<0.05$, |log2FC|>1).

Weighted Gene Co-Expression Network Analysis (WGCNA) (*Langfelder and Horvath, 2008*) was used to analyse the correlation between genes and phenotypes. We selected an appropriate 'soft thresholding power' using the 'picksoftthreshold' function in the WGCNA package (v1.61). Next, 'blockwiseModules' function was used to construct the co-expression matrix with the option 'check-MissingData = TRUE, power = 16, TOMType='unsigned,' minModuleSize = 30, maxBlockSize = 6000, mergeCutHeight = 0.25.' The modules with high correlation to T2DM phenotype and $p<0.05$ were selected for downstream analysis, and genes in these modules were used for GO and KEGG functional enrichment analysis. Hub genes were identified based on the module eigengene-based connectivity (kME), |kME>0.8| as the cut-off criteria.

Instead of identifying the differentially expressed genes within a pathway between the two groups, Aggregate Fold Change (AFC) calculated the average multiple change for each gene and defined the pathway score as the average difference multiple for all genes in that pathway. A null hypothesis

test was performed using the pathway scores of the gene expression dataset, and the significance of each pathway was estimated by p-value (*Yu et al., 2017*). STRING (v11.0) provides a tool for functional enrichment analysis based on AFC. GO and KEGG enrichment analysis was performed using g: Profiler (*Raudvere et al., 2019*), AFC enrichment analysis was performed using STRING (https://string-db.org/).

## Untargeted metabolomics processing

The 200 µL homogenized fecal sample was mixed with 800 µL cold methanol/acetonitrile (1:1, v/v) to remove the protein. The mixture was centrifuged for 15 min (14,000 g, 4 °C) followed by the drying of the supernatant in a vacuum centrifuge. For LC-MS analysis, the samples were re-dissolved in 100 µL acetonitrile/water (1:1, v/v) solvent. LC-MS/MS analysis was performed using an UHPLC (1290 Infinity LC, Agilent Technologies) coupled to a quadrupole time-of-flight (AB Sciex TripleTOF 6600) in Shanghai Applied Protein Technology Co., Ltd. The samples were separated by Agilent 1290 Infinity LC ultra-high performance liquid chromatography (UHPLC) HILIC column. In both ESI positive and negative modes, the mobile phase contained A=25 mM ammonium acetate and 25 mM ammonium hydroxide in water and B=acetonitrile. The gradient was 85% B for 1 min and was linearly reduced to 65% in 11 min, and then was reduced to 40% in 0.1 min and kept for 4 min, and then increased to 85% in 0.1 min, with a 5 min re-equilibration period employed. AB Triple TOF 6600 mass spectrometer was used to collect the primary and secondary spectra of per sample. After separation by UHPLC, the samples were analyzed by Triple TOF 6600 mass spectrometer (AB SCIEX). Positive and negative modes of electrospray ionization (ESI) were, respectively, detected.

## Targeted medium- and long-chain fatty acid metabolomics processing

A total amount of 100 µL plasma per sample was taken in 2 mL glass centrifuge tubes and 1 mL chloroform-methanol solution was added. After 30 min of ultrasound, the supernatant was taken and 2 mL of 1% sulfuric acid-methanol solution was added. The mixed solution was placed in a water bath at 80°C, for 30 min, and then 1 mL of N-hexane and 5 mL of pure water were added in turn. Next, 500 µL of supernatant was absorbed, 25 µL of methyl N-nonaconate was added and mixed. The final sample was detected by GC-MS. All samples were separated by Agilent DB-WAX capillary column (30 m×0.25 mm ID ×0.25 µm) gas chromatography. Agilent 7890/5975 C gas-mass spectrometer was used for mass spectrometry. The chromatographic peak area and retention time were extracted by MSD ChemStation software. The content of medium-and long-chain fatty acids was calculated by drawing a standard curve.

## Metabolomics statistical processing

The screening of significant changed metabolites was performed using univariate and multidimensional analysis. Student's t-test was applied to determine the significance of differences between two groups. The variable importance in the projection (VIP) value of each variable was obtained from the Orthogonal Partial Least Squares Discriminant Analysis (OPLS-DA) model, which was used to indicate its contribution to the classification. The screening criteria were $p < 0.05$ and VIP $> 1$.

## Animal treatments

Specific-pathogen-free C57BL/6 J male mice at 6 weeks of age (vendor: Chengdu Dossy Experimental Animals Co., LTD) were randomly assigned to four groups (FTPA, FT, PA, and Control). All the mice lived in cages with the same conditions, including 12 hr light and 12 hr dark cycles, temperature 22–25°C and humidity 40–60%.

## Diets

HPAD was prepared by adding 12% PA to conventional forage. Both the conventional forage and the HPAD were sterile and the fresh forage was renewed three times a week. On day 0, the diets of FTPA-treated mice and PA-treated mice were switched to HPAD, FT-treated mice and control mice were still fed with conventional forage. Drinking water was sterile and renewed twice a day.

## Transplant preparation and use

After single-cage feeding, FPG detection and fecal collection were performed, and fecal samples of seven T2DM macaques were mixed for the preparation of transplants. The appropriate volume of

diluent was added to the fecal sample (i.e. add 4 ml diluent per gram of feces) and the preparation of diluent can be found in *Berland et al., 2021*. Sodium L-ascorbic acid and L-cysteine hydrochloride monohydrate were added to all suspensions at final concentrations of 5% (w/v) and 0.1% (w/v), respectively (The sterile diluent of control group was added with the same amount of reagent). The mixture was homogenized and filtered with a 200-mesh sterile mesh screen to remove large particles from the feces, and the filtrate was passed through 400 and 800 sterile mesh screens to remove undigested food and smaller particulate matter. The filtrate was divided into 50 ml centrifuge tubes, centrifuged at 600×g for 5 min, and the precipitation was discarded. Finally, the fecal supernatant was divided into new centrifuge tubes in equal parts (400 μL per tube) and frozen at –80 °C. For use, the transplant was quickly thawed in a 37°C water bath.

### Fecal microbiota transplantation (FMT)

After 1 week of changed feeding regime, the FTPA and FT mice groups were pre-treated with 1 g/L neomycin sulfate, 1 g/L ampicillin, and 1 g/L metronidazole in the drinking water for 14 days, and the control group was not treated. For FMT treatment, the gavage with 400 μL transplant, which were thawed ahead of time, was performed for 14 days in FTPA and FT-treated mice. At the same time, the gavage in control and PA-treated mice were performed with sterile diluent.

### Metabolic measurements

Throughout the experiment, body weight and feces were collected every month, FPG was detected every half month under fasting for at least 12 hr. OGTT was performed on day 110 and ITT was performed on day 115. For OGTT, after a 12 hr overnight fast, oral glucose gavage (1.2 g/kg of 12% dextrose solution) was performed and followed by blood sample collection from the tail vein at 0, 15, 30, 60, 90, and 120 min. For ITT, after a 6 hr fast, intraperitoneal insulin injection was performed (0.75 U/kg, human regular insulin), followed by the blood samples collection from the tail vein at 0, 20, 40, 60, 90, and 120 min. On day 120, after FPG detection, blood collection and issues were collected for quantitative RT-PCR, H-E staining, and ELISA.

### Isolation of tissue for quantitative RT-PCR and ELISA

Tissue was homogenized for RNA extraction, after adjusting the final concentration of RNA, and DNA reverse transcription was performed. Quantitative RT-PCR was performed on CFX Connect Real-Time PCR Detection System (Bio-Rad, USA) using SYBR Green (*Tables 3–5*). PA, IL-1β, IL-6, TNF-α, and IL-17A were detected using the Jiangsu Meimian ELISA kit and followed the operator instructions.

### H-E staining

The isolated livers and pancreas were dehydrated and embedded after fixation with formalin for 24 hr. Paraffin-embedded livers and pancreas specimens were cut at a thickness of 3 μm. All sections were stained with hematoxylin then eosin, and finally, microscopy and image acquisition were performed.

### Statistical analysis

In this study, one-way ANOVA was used to determine statistical significance for comparisons of more than three groups, and for comparisons of two groups, two-tailed t-test was used. P-values are represented on figures as follows: ns, not significant, $*p < 0.05$, $**p < 0.01$.

## Acknowledgements

This work was supported by the Science and Technology Foundation of Sichuan Province (2021YJ0136) and the National Natural Science Foundation of China (No. 32171607). Special thanks to Sichuan Greenhouse Biotech Co., Ltd. for the sample collection and thanks to Prof. Jinchuan Xing and Dr. Megan Price for revising the manuscript.

## Additional information

### Funding

| Funder | Grant reference number | Author |
|---|---|---|
| Science and Technology Foundation of Sichuan Province | 2021YJ0136 | Jing Li |
| National Natural Science Foundation of China | 32171607 | Jing Li |

The funders had no role in study design, data collection and interpretation, or the decision to submit the work for publication.

### Author contributions

Xu Liu, Conceptualization, Data curation, Software, Formal analysis, Validation, Investigation, Visualization, Methodology, Writing - original draft, Project administration, Writing - review and editing; Yuchen Xie, Data curation, Software, Formal analysis, Validation, Investigation, Visualization, Writing - review and editing; Shengzhi Yang, Conceptualization, Data curation, Software, Formal analysis, Methodology; Cong Jiang, Conceptualization, Data curation, Investigation; Ke Shang, Jinxia Luo, Lin Zhang, Validation, Investigation, Methodology; Gang Hu, Qinghua Liu, Resources, Funding acquisition, Project administration; Bisong Yue, Jing Li, Conceptualization, Resources, Supervision, Funding acquisition, Methodology, Project administration, Writing - review and editing; Zhenxin Fan, Conceptualization, Supervision, Project administration, Writing - review and editing; Zhanlong He, Supervision, Project administration, Writing - review and editing

### Author ORCIDs

Zhenxin Fan (ID) https://orcid.org/0000-0003-0422-9497
Jing Li (ID) https://orcid.org/0000-0002-3242-4495

### Ethics

This study followed animal welfare guidelines throughout the sample collection process, and all observations and samplings were approved by the Sichuan University's Animal Care Committee (Permit Number: SCU210429001 and SCU230810001).

Reviewer #1 (Public review): https://doi.org/10.7554/eLife.104355.4.sa1
Author response https://doi.org/10.7554/eLife.104355.4.sa2

---

## Additional files

### Supplementary files

Supplementary file 1. Information on the macaque samples.
Supplementary file 2. List of 64 differential metabolites.
Supplementary file 3. List of genes in four modules significantly correlated with T2DM.
Supplementary file 4. List of all identified untargeted metabolites.
Supplementary file 5. List of all targeted medium- and long-chain fatty acid metabolites.
MDAR checklist

### Data availability

The raw data of transcriptomes, metagenomes, and 16S rRNA have been submitted to the China National Center for Bioinformation/Beijing Institute of Genomics, Chinese Academy of Sciences with BioProject accession no. PRJCA021499. Transcriptomes data: (GSA: CRA013604). 16S rRNA data (macaque): (GSA: CRA013637). 16S rRNA data (mouse): (GSA: CRA013638). Metagenomes data: (GSA: CRA013607). The identified untargeted metabolites are listed in *Supplementary file 4*. The identified targeted medium-and long-chain fatty acid metabolites are listed in *Supplementary file 5*.

The following datasets were generated:

| Author(s) | Year | Dataset title | Dataset URL | Database and Identifier |
|---|---|---|---|---|
| Liu X | 2024 | Multi-omics investigation of spontaneous T2DM macaque reveals gut microbiota promote T2DM by up-regulating the absorption of excess palmitic acid | https://ngdc.cncb.ac.cn/bioproject/browse/PRJCA021499 | NGDC BioProject, PRJCA021499 |
| Liu X | 2025 | Multi-omics investigation of spontaneous T2DM macaque reveals gut microbiota promotes T2DM by up-regulating the absorption of excess palmitic acid | https://ngdc.cncb.ac.cn/gsa/browse/CRA013604 | Genome Sequence Archive, CRA013604 |
| Liu X | 2025 | macaque gut 16S rRNA | https://ngdc.cncb.ac.cn/gsa/browse/CRA013637 | Genome Sequence Archive, CRA013637 |
| Liu X | 2025 | mouse gut 16S rRNA | https://ngdc.cncb.ac.cn/gsa/browse/CRA013638 | Genome Sequence Archive, CRA013638 |
| Liu X | 2025 | macaque gut metagenome | https://ngdc.cncb.ac.cn/gsa/browse/CRA013607 | Genome Sequence Archive, CRA013607 |

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
