## [Editor Report · eLife Assessment]

This **important** work substantially advances our understanding of the interaction among gut microbiota, lipid metabolism, and the host in type 2 diabetes. The evidence supporting the claims of the authors is **convincing**. The work will be of interest to medical biologists working on microbiota and diabetes.

---

## [Referee Report · Reviewer #1 (Public review)]

Summary:

The authors sought to identify the relationships between gut microbiota, lipid metabolites and the host in type 2 diabetes (T2DM) by using spontaneously developed T2DM in macaques, considered among the best human models.

Strengths:

The authors compared comprehensively the gut microbiota, plasma fatty acids between spontaneous T2DM and the control macaques, verifying the results with macaques in a high-fat diet-fed mice model.

Comments on revisions:

The authors responded to the comments raised, and the manuscript has been improved.

---

## [Author Response]

The following is the authors’ response to the previous reviews

**Reviewer #1 (Public review):**
Summary:The authors tried to identify the relationships between gut microbiota, lipid metabolites and the host in type 2 diabetes (T2DM) by using spontaneously developed T2DM in macaques, considered among the best human models.Strengths:The authors compared comprehensively the gut microbiota, plasma fatty acids between spontaneous T2DM and the control macaques, and tried verified the results with macaques in high-fat diet-fed mice model.Weaknesses:The observed multi-omics on macaques can be done on humans, which weakens the conclusion of the manuscript, unless the observation/data on macaques could cover during the onset of T2DM that would be difficult to obtain from humans.Regarding the metabolomic analysis on fatty acids, the authors did not include the results obtained form the macaque fecal samples which should be important considering the authors claimed the importance of gut microbiota in the pathogenesis of T2DM. Instead, the authors measured palmitic acid in the mouse model and tried to validate their conclusions with that.In murine experiments, palmitic acid-containing diet were fed to mice to induce diabetic condition, but this does not mimic spontaneous T2DM in macaques, since the authors did not measure in macaque feces (or at least did not show the data from macaque feces of) palmitic acid or other fatty acids; instead, they assumed from blood metabolome data that palmitic acid would be absorbed from the intestine to affect the host metabolism, and added palmitic acid in the diet in mouse experiments. Here involves the probable leap of logic to support their conclusions and title of the study.In addition, the authors measured omics data after, but not before, the onset of spontaneous T2DM of macaques. This can reveal microbiota dysbiosis driven purely by disease progression, but does not support the causative effect of gut microbiota on T2DM development that the authors claims.

We are sorry for misunderstanding your point and failing to address your question regarding macaque fecal metabolomics in our previous response. Our study performed untargeted metabolomics on macaque feces and indeed detected the differential metabolite palmitic acid (PA) content, which showed an obvious decrease in T2DM macaques compared with the control (Table 1). However, the difference in PA level between the two groups was not significant (p = 0.17). It may be attributed to the limitation of untargeted metabolomics methodology in absolute quantitative analysis. In addition, we found many other long-chain fatty acids were down-regulated in the T2DM macaque feces (Table 1). Such results are consistent with our observation in murine experiments. We examined PA levels in the feces, ileum, and serum in mice and found that PA level was significantly decreased in fecal samples but increased in the ileum and serum. These findings demonstrated that without the transplantation of gut microbiota, the ileum could not absorb the PA effectively even at a high concentration of ingested PA. Only mice receiving fecal microbiota transplants from T2DM macaques and fed a high-PA diet showed a significant increase in the ileum and serum alongside a decrease in fecal PA concentration. Both the macaque metabolomics and mice experiment results suggest that gut microbiota mediated the absorption of excess PA in the ileum leading to the accumulation of PA in the serum. In the revised manuscript, we added the results of all differential metabolites in Table S2.

**Author response table 1. sa2table1:** Differential analysis of palmitic acid and other fatty acids from fecal untargeted metabolomics in macaques.

SuperClass	Class	SubClass	Names	log2FoldChange	p-value	VIP	Type	Significant
Lipids and lipid-like molecules	Fatty Acyls	Fatty acids and conjugates	Palmitic acid	–0.449600	0.173730	0.881851	Down	no
Lipids and lipid-like molecules	Fatty Acyls	Fatty acids and conjugates	2-hydroxypalmitic acid	–0.239318	0.408640	0.526705	Down	no
Lipids and lipid-like molecules	Fatty Acyls	Fatty acids and conjugates	2-isopropylmalic acid	–0.651460	0.210910	1.348017	Down	no
Lipids and lipid-like molecules	Fatty Acyls	Fatty acids and conjugates	Adipic acid	–0.621198	0.265750	1.385746	Down	no
Lipids and lipid-like molecules	Fatty Acyls	Fatty acids and conjugates	Caproic acid	–1.115620	0.127570	1.625076	Down	no
Lipids and lipid-like molecules	Fatty Acyls	Fatty acids and conjugates	12(13)-epoxy-9z-octadecenoic acid	–1.091521	0.905650	0.418656	Down	no
Lipids and lipid-like molecules	Fatty Acyls	Fatty acids and conjugates	12s-hydroxy-5z,8e,10e-heptadecatrienoic acid	–1.489847	0.227590	1.278136	Down	no
Lipids and lipid-like molecules	Fatty Acyls	Fatty acids and conjugates	18-carboxydinorleukotriene b4	–0.401997	0.214310	1.193911	Down	no
Lipids and lipid-like molecules	Fatty Acyls	Fatty acids and conjugates	5,8,11,14-eicosatetraynoic acid	–0.422578	0.250660	1.064897	Down	no
Lipids and lipid-like molecules	Fatty Acyls	Fatty acids and conjugates	9-oxo-11-(3-pentyl-2-oxiranyl)–10e-undecenoic acid	–0.094902	0.670110	0.434639	Down	no
Lipids and lipid-like molecules	Fatty Acyls	Fatty acids and conjugates	Behenic acid	–0.842212	0.649980	0.367039	Down	no
Lipids and lipid-like molecules	Fatty Acyls	Fatty acids and conjugates	Cis-9-palmitoleic acid	–0.186967	0.560480	0.571173	Down	no
Lipids and lipid-like molecules	Fatty Acyls	Fatty acids and conjugates	Dodeca-2(e),4(e)-dienoic acid	–0.131473	0.901130	0.033862	Down	no
Lipids and lipid-like molecules	Fatty Acyls	Fatty acids and conjugates	Fumagillin	–0.105898	0.353840	0.952419	Down	no
Lipids and lipid-like molecules	Fatty Acyls	Fatty acids and conjugates	Hymeglusin	–0.625361	0.030932	2.087180	Down	yes
Lipids and lipid-like molecules	Fatty Acyls	Fatty acids and conjugates	Lignoceric acid	–1.247611	0.062477	1.858818	Down	no
Lipids and lipid-like molecules	Fatty Acyls	Fatty acids and conjugates	Maresin 1	–0.339035	0.950420	0.129840	Down	no
Lipids and lipid-like molecules	Fatty Acyls	Fatty acids and conjugates	Monensin	–1.646496	0.021254	2.263949	Down	yes
Lipids and lipid-like molecules	Fatty Acyls	Fatty acids and conjugates	Myriocin	–0.915527	0.023194	2.149777	Down	yes
Lipids and lipid-like molecules	Fatty Acyls	Fatty acids and conjugates	Palmitic acid alkyne	–0.137112	0.937670	0.084572	Down	no
Lipids and lipid-like molecules	Fatty Acyls	Fatty acids and conjugates	Tetradecanedioic acid	–0.403117	0.523220	0.602053	Down	no
Lipids and lipid-like molecules	Fatty Acyls	Fatty acids and conjugates	Trans-traumatic acid	–0.208456	0.855200	0.314711	Down	no
Lipids and lipid-like molecules	Fatty Acyls	Fatty acids and conjugates	Traumatic Acid	–0.382543	0.436360	0.696223	Down	no
Lipids and lipid-like molecules	Fatty Acyls	Fatty acids and conjugates	10-hydroxy-4z,7z,11e,13z,16z,19z-docosahexaenoic acid	–0.295357	0.290920	0.589113	Down	no
Lipids and lipid-like molecules	Fatty Acyls	Fatty acids and conjugates	11(12)-epoxy-5z,8z,14z,17z- eicosatetraenoic acid	–0.421126	0.459440	0.431261	Down	no
Lipids and lipid-like molecules	Fatty Acyls	Fatty acids and conjugates	12,13-dihydroxy-9z-octadecenoic acid	–1.091521	0.905650	0.418656	Down	no
Lipids and lipid-like molecules	Fatty Acyls	Fatty acids and conjugates	13-hydroxy-4z,7z,10z,14e,16z,19z-docosahexaenoic acid	–0.061835	0.825750	0.098187	Down	no
Lipids and lipid-like molecules	Fatty Acyls	Fatty acids and conjugates	2-hydroxy-3-methylbutyric acid	–0.637274	0.522920	0.475814	Down	no
Lipids and lipid-like molecules	Fatty Acyls	Fatty acids and conjugates	9-deoxy-9-methylene-16,16-dimethylprostaglandin e2	–0.107350	0.932850	0.126214	Down	no
Lipids and lipid-like molecules	Fatty Acyls	Fatty acids and conjugates	.alpha.-keto-.gamma.-(methylthio)butyric acid	–0.797116	0.044712	1.973034	Down	yes
Lipids and lipid-like molecules	Fatty Acyls	Fatty acids and conjugates	Azelaic acid	–0.250671	0.561220	0.516314	Down	no
Lipids and lipid-like molecules	Fatty Acyls	Fatty acids and conjugates	Butanoic acid	–0.389470	0.408450	0.663941	Down	no
Lipids and lipid-like molecules	Fatty Acyls	Fatty acids and conjugates	Cis,cis-muconic acid	–0.165996	0.907660	0.662026	Down	no
Lipids and lipid-like molecules	Fatty Acyls	Fatty acids and conjugates	Dodecanedioic acid	–0.005762	0.783380	0.259046	Down	no
Lipids and lipid-like molecules	Fatty Acyls	Fatty acids and conjugates	Dodecanoic acid	–0.674704	0.353100	0.982481	Down	no
Lipids and lipid-like molecules	Fatty Acyls	Fatty acids and conjugates	Erucic acid	–0.236100	0.322980	0.672230	Down	no
Lipids and lipid-like molecules	Fatty Acyls	Fatty acids and conjugates	Fa 18:1+3o	–0.080505	0.818690	0.244925	Down	no
Lipids and lipid-like molecules	Fatty Acyls	Fatty acids and conjugates	Fahfa 34:0	–0.284621	0.254680	0.680759	Down	no
Lipids and lipid-like molecules	Fatty Acyls	Fatty acids and conjugates	Fahfa 36:1	–0.165777	0.287690	0.657585	Down	no
Lipids and lipid-like molecules	Fatty Acyls	Fatty acids and conjugates	Heptadecanoic acid	–0.686373	0.147130	0.880272	Down	no
Lipids and lipid-like molecules	Fatty Acyls	Fatty acids and conjugates	Hexadecanedioic acid, 3,3,14,14-tetramethyl-	–0.166211	0.586710	0.156649	Down	no
Lipids and lipid-like molecules	Fatty Acyls	Fatty acids and conjugates	Hydroxyisocaproic acid	–1.255793	0.322530	1.018523	Down	no
Lipids and lipid-like molecules	Fatty Acyls	Fatty acids and conjugates	Isovaleric acid	–0.466135	0.217710	1.330000	Down	no
Lipids and lipid-like molecules	Fatty Acyls	Fatty acids and conjugates	Mevalonic acid	–0.025494	0.669590	0.665701	Down	no
Lipids and lipid-like molecules	Fatty Acyls	Fatty acids and conjugates	Myristic acid	–0.732180	0.156680	1.349036	Down	no
Lipids and lipid-like molecules	Fatty Acyls	Fatty acids and conjugates	Octadecanedioic acid	–1.376197	0.105400	1.407770	Down	no
Lipids and lipid-like molecules	Fatty Acyls	Fatty acids and conjugates	Octadecanoic acid	–0.172547	0.927540	0.308819	Down	no
Lipids and lipid-like molecules	Fatty Acyls	Fatty acids and conjugates	Pentadecanoic acid	–0.367656	0.329110	0.535937	Down	no
Lipids and lipid-like molecules	Fatty Acyls	Fatty acids and conjugates	Pimelic acid	–1.036139	0.041123	1.997825	Down	yes
Lipids and lipid-like molecules	Fatty Acyls	Fatty acids and conjugates	Ricinoleic acid	–0.354777	0.678380	0.587595	Down	no
Lipids and lipid-like molecules	Fatty Acyls	Fatty acids and conjugates	Sebacic acid	–0.278685	0.460530	0.460984	Down	no
Lipids and lipid-like molecules	Fatty Acyls	Fatty acids and conjugates	Tridecanoic acid (Tridecylic acid)	–1.592480	0.008072	2.035876	Down	yes
Lipids and lipid-like molecules	Fatty Acyls	Fatty acids and conjugates	(z)–5,8,11-trihydroxyoctadec-9-enoic acid	–0.026995	0.782280	0.161350	Down	no
Lipids and lipid-like molecules	Fatty Acyls	Fatty acids and conjugates	5-heptenoic acid, 7-[(1 r,2r,3s,5s)–2-[(1e,3s)–3-(2,3-dihydro-1h-inden-2-yl)–3-hydroxy-1-propen-1-yl]–3-fluoro-5-hydroxycyclopentyl]-, (5z)-	–0.150346	0.879000	0.246002	Down	no

Regarding the causative effect of gut microbiota on T2DM development, we agree with the reviewer that the omics data were obtained after, but not before, the onset of spontaneous T2DM macaques, the microbiota dysbiosis is probably driven by disease progression. For this reason, we have changed the title of our manuscript and some of our conclusions, which can be found in our response below.

**Reviewer #1 (Recommendations for the authors):**
As described above, the data presented does not support the notion that gut microbiota change in T2DM macaques promote the disease - rather it showed the outcome of the disease progression. In addition, the involvement of palmitic acid absorption was only shown in mice but not in macaques. Therefore, the authors should change their title and conclusions to more precisely reflect their observation.

According to your suggestion, we changed the title and the conclusion to make them more precise and avoid emphasizing the causative effect of gut microbiota on T2DM. The new title is “Multi-omics investigation of spontaneous T2DM macaque emphasizes gut microbiota could up-regulate the absorption of excess palmitic acid in the T2DM progression”. We also revised the wording of the results and conclusions to acknowledge the limitation of our study, “We also revealed the specific structure of gut microbiota that promoted T2DM development by regulating the absorption of excess PA in mice, providing experimental evidence for the functional role of gut microbiota in T2DM pathogenesis.” (Lines 122-125), “In particular, concentrations of PA, palmitoleic acid, and oleic acid were significantly higher in the T2DM group than control group (p<0.05 and VIP>1). The concentration of PA in the plasma of T2DM macaques increased, while the concentration of palmitic acid in the feces decreased (Figures 3F and G, Table S2).” (Lines 228-233), and “Our study confirms the functional role of gut microbiota and PA in the T2DM progression. The microbiota composition, specifically higher abundance of R. gnavus (current name: M. gnavus) and Coprococcus sp., and lower abundance of Treponema, F. succinogenes, Christensenellaceae, and F16, promoted the absorption of excess PA which is important for the development of T2DM. However, in this study, such microbial alterations were detected in macaques after they had developed the disease of T2DM instead of before or onset of T2DM, the causative effect of gut microbiota and their action mechanism on the development of T2DM is worth further investigation.” (Lines 450-458).